# Ceramic Nanomaterials in Caries Prevention: A Narrative Review

**DOI:** 10.3390/nano12244416

**Published:** 2022-12-11

**Authors:** Mohammed Zahedul Islam Nizami, Veena Wenqing Xu, Iris Xiaoxue Yin, Christie Ying Kei Lung, John Yun Niu, Chun Hung Chu

**Affiliations:** Faculty of Dentistry, University of Hong Kong, Hong Kong SAR 999077, China

**Keywords:** caries, nanoparticles, nanomaterials, remineralising, dentin, dentistry, prevention

## Abstract

Ceramic nanomaterials are nanoscale inorganic metalloid solids that can be synthesised by heating at high temperatures followed by rapid cooling. Since the first nanoceramics were developed in the 1980s, ceramic nanomaterials have rapidly become one of the core nanomaterials for research because of their versatility in application and use in technology. Researchers are developing ceramic nanomaterials for dental use because ceramic nanoparticles are more stable and cheaper in production than metallic nanoparticles. Ceramic nanomaterials can be used to prevent dental caries because some of them have mineralising properties to promote the remineralisation of tooth tissue. Ceramic minerals facilitate the remineralisation process and maintain an equilibrium in pH levels to maintain tooth integrity. In addition, ceramic nanomaterials have antibacterial properties to inhibit the growth of cariogenic biofilm. Researchers have developed antimicrobial nanoparticles, conjugated ceramic minerals with antibacterial and mineralising properties, to prevent the formation and progression of caries. Common ceramic nanomaterials developed for caries prevention include calcium-based (including hydroxyapatite-based), bioactive glass-based, and silica-based nanoparticles. Calcium-based ceramic nanomaterials can substitute for the lost hydroxyapatite by depositing calcium ions. Bioactive glass-based nanoparticles contain surface-reactive glass that can form apatite crystals resembling bone and tooth tissue and exhibit chemical bonding to the bone and tooth tissue. Silica-based nanoparticles contain silica for collagen infiltration and enhancing heterogeneous mineralisation of the dentin collagen matrix. In summary, ceramic nanomaterials can be used for caries prevention because of their antibacterial and mineralising properties. This study gives an overview of ceramic nanomaterials for the prevention of dental caries.

## 1. Introduction

Dental caries is the mineral loss of dental hard tissues (enamel, dentin, cementum) caused by fermentable acid produced by cariogenic bacteria. Thus, altering the oral microenvironment to an acidic environment can cause hard tissue demineralisation in a period and lead to tooth decay or caries (Figure 1) [1]. However, caries is a preventable disease, and early caries can be remineralised under a favourable environment [2]. Inhibiting cariogenic bacteria and biofilm or enhancing remineralisation, or applying dual action, can be a scientific approach for preventing the initiation of primary caries. Fluoride has been the first attempt in dental practice used for preventive purposes [3]. Subsequently, casein phosphopeptide-amorphous calcium phosphate [4] has been recently introduced and has shown promising results. Nowadays, researchers are using nanotechnology to develop multifunctional nanomaterials for preventing caries [5]. An ideal remineralising agent should transport remineralising minerals and ions to the deeper surface of the carious enamel or dentin to promote deep remineralisation. Moreover, materials should have antimicrobial properties against cariogenic microbes [6]. They protect teeth from demineralisation without making bacteria resistant. Hence, the strategies for caries prevention are microbe inhibition and remineralisation enhancement. Nanotechnology is a recent research trend and has been investigated to develop anticaries materials. They exhibit unique physical, chemical, and biological properties, such as large surface-to-volume ratios. There are arguments for the development of novel nanomaterials for caries prevention [7].

Ceramic nanomaterials are nanoscale materials that are inorganic metalloid solids made up of oxides, carbides, carbonates, and phosphates. These ceramic nanomaterials are synthesised by heating at high temperatures followed by rapid cooling [9]. Ceramic nanoparticles are more stable and cheaper in production than metallic nanoparticles. Most ceramic nanoparticles resemble tooth minerals (hydroxyapatite, also known as a calcium phosphate ceramic). Ceramic materials are biocompatible and have a high affinity to tooth structure. In addition, they exhibit antimicrobial properties against cariogenic microbes [10]. Common ceramic nanomaterials studied for caries prevention include calcium-based (including hydroxyapatite-based), bioactive glass-based, and silica-based nanoparticles. They could be potential candidates for caries prevention. In this paper, we comprehensively review ceramic nanomaterials that have been studied for caries prevention.

## 2. Method

Two investigators searched publications in English on ceramic nanomaterials (including their nanocomposite) for caries prevention. They searched three databases: PubMed, EMBASE, and Web of Science. The keywords were (nanomaterials OR nanoparticles OR nanocomposites) AND (caries OR tooth decay OR demineralisation OR remineralisation). The search was restricted to publications in English. No publication year limit was set. The last search was performed on 11 November 2022 (Figure 2).

The two investigators removed duplicate publications to attain a list of publications. They screened the titles and abstracts to exclude literature reviews, abstracts, publications not related to dental caries or ceramic nanomaterials, publications on ceramic nanomaterials that were not in nanoscale, and other irrelevant publications. The two investigators retrieved the full texts of the remaining publications for review. They then performed a manual screening of the reference lists in the selected publications. They discussed the selected publications with another investigator to achieve an agreement on the list of publications included in this review.

## 3. Result

The initial literature search revealed 2318 publications (840 articles in PubMed, 441 articles in EMBASE, and 1037 articles in Web of Science). In total, 683 duplicate publications were removed. After screening the titles and abstracts, 1572 publications were removed, as they were literature reviews, abstracts, publications not related to dental caries or ceramic nanomaterials, publications on ceramic nanomaterials that were not in nanoscale, or other irrelevant publications. The references of these selected 63 publications were searched, and 17 publications that met the inclusion criteria were added. A total of 80 publications met the eligibility criteria and were included in this review.

Based on the included publications, the investigators categorised ceramic nanomaterials into *calcium-based nanoparticles (including hydroxyapatite-based nanoparticles), bioactive glass-based nanoparticles,* and *silica-based nanoparticles* (Table 1).

## 4. Discussion

### 4.1. Mechanisms of Caries Progression and Prevention

The mechanisms behind dental caries are well-established. Studies have commonly described microbial (cariogenic bacteria and biofilm) effects and dental hard tissue demineralisation for the caries mechanism. However, due to the complex nature of caries progression, these mechanisms are not linear. Cariogenic bacteria grow on surfaces as organised groups called dental plaque. This dental plaque is basically a biofilm. Biofilm leads to caries formation. However, precepted biofilm on a tooth surface does not confirm the presence of caries. Caries initiates only after a complex interaction of host factors, including stagnation area (the tooth surface), fermentable carbohydrate (free sugars), and cariogenic bacteria that can lead to caries expression over time (Figure 3) [95].

Organic acids produced by biofilm bacteria demineralise the crystalline mineral structure of the tooth in demineralisation. Lactic acid predominantly exists in this process and is considered to be the main acid involved in caries formation [96]. In this acidic condition, the pH level drops to a favourable condition for the dissolution of hydroxyapatite in dental hard tissue. Hydrogen ions in the acidic environment dissolve hydroxyapatite, producing calcium ions, phosphate ions, and water. Therefore, the surface demineralisation of the tooth occurs [97]. After that, the loss of minerals leads to developing permeability and porosity, enamel crystal derangement, and further acid diffusion to enamel pores. This acid diffusion decreases the pH around the enamel crystals and further dissolves the hydroxyapatite [98]. Some of the anticaries agents inhibit the growth of cariogenic bacteria to decrease the organic acids produced by bacteria. Some of the anticaries agents protect the surface layer from further demineralisation and facilitate remineralisation when the calcium and phosphate content increases in saliva [99].

The buffering of saliva plays a crucial role in maintaining a neutral pH in the oral environment (Figure 3). The increased pH value makes saturated calcium and phosphate ions redeposited, leading demineralisation to stop and minerals to add back to the dissolved enamel surface. Therefore, partially dissolved enamel crystal and enamel surface become remineralised. Saliva is essential for this remineralisation and maintaining tooth integrity. Saliva can be supplemented with an antibacterial/antibiofilm component or remineralising mineral components or their combination to prevent caries formation and progression. It can arrest demineralisation and facilitate remineralisation. In Figure 4, the schematic illustration explains the mechanism of de- and remineralisation.

### 4.2. Calcium-Based Nanoparticles

Dietary sources provide calcium, which is an essential mineral for teeth and bones [40]. Unlike acidic desolation, the leak of calcium also starts the demineralisation of the tooth, resulting in dental caries [100]. Calcium-enriched saliva facilitates remineralisation. Calcium and phosphate ions are mainly responsible for the inhibition of demineralisation and enhancement of remineralisation and act as a natural defence mechanism against dental caries. However, in the persistent cariogenic condition, the balance of these ions disrupts and rearranges the enamel surface [1]. A secondary supply can overcome the requirement. Noninvasive caries management by remineralisation has been shown to be a major advantage in clinical dentistry. Therefore, some researchers have employed calcium nanomaterials to meet these requirements. Recently, researchers have studied several calcium phosphate-based remineralisation systems for caries management [101].

Hydroxyapatite is a natural mineral in the form of a calcium phosphate apatite that is similar to the human hard tissues in morphology and chemical composition [102]. Hydroxyapatite is the key component of teeth and bones. It usually exists with a length of 60 nm and a width of 5–20 nm. It is responsible for the rigidity and strength of the basic structure of hard dental tissue [103]. Nanohydroxyapatite has received great attention and is promising in cariology research for its morphological and mineral structure similarity with bone and teeth [104]. Due to its biocompatibility, bioactivity, and antibacterial effect [12], it can enhance several beneficial properties of existing restorative materials [105].

Several studies have reported the remineralisation potentials of nanohydroxyapatite when researchers incorporated it into restorative materials or toothpaste [14,24,106]. The efficacy of nanohydroxyapatite in remineralising caries lesions was effective [11]. A study examined different nanohydroxyapatite concentrations on initial enamel caries lesions under dynamic pH-cycling conditions and found that nanohydroxyapatite improved surface microhardness [12]. It was also reported that nanohydroxyapatite particles were deposited on the cellular structure of the demineralised enamel and formed new layers on the enamel surface.

Simultaneously, a study found that the remineralisation effect of nanohydroxyapatite on demineralised bovine enamel was better than that of microhydroxyapatite in different pH cycling conditions [13]. In addition, the researchers described that nanohydroxyapatite can contribute to both the particle and ion-regulated remineralisation for repairing demineralised enamel.

At the same time, a study added nanohydroxyapatite to a sports drink and found it to be effective against dental erosion [25]. In another study, nanohydroxyapatite-treated enamel block showed a protective layer formation with increased microhardness on the enamel surface in a cariogenic condition [27].

One study reported higher remineralising effects of nanohydroxyapatite compared to amine fluoride toothpaste on bovine dentine and suggested using nanohydroxyapatite for caries prevention [14]. At the same time, an article investigated nanohydroxyapatite and *Galla chinensis* on the remineralisation of initial enamel caries lesions [26]. The researchers reported that they had found enhanced remineralisation by depositing more minerals to the decay to reduce the depth of the lesions.

Another study reported the remineralisation effect of nanohydroxyapatite in a conjugate of sealant and found it effective in sealing demineralised microleakage of enamel pits and fissures by depositing nanoparticles [28]. It also maintained the shear bond strength of the sealant. Thus, the study suggested use in minimal intervention dentistry applications for sealing demineralised pits and fissures on the enamel.

Another study showed no demineralisation in the sound enamel when it is exposed to nanohydroxyapatite [15]. These researchers demonstrated that nanohydroxyapatite dentifrice showed remineralisation comparable to fluoride. Thus, they suggested using nanohydroxyapatite as an alternative to fluoride toothpaste to prevent caries. In another article, researchers employed nanohydroxyapatite gel, ozone therapy, and their combination therapy [16]. They reported that although these exert some capacities for remineralisation individually, the combination showed the best effect in nonrestorative caries management. Therefore, the researchers suggested using this combination therapy for a longer period to provide nonrestorative caries treatment.

Another study showed that nanohydroxyapatite toothpaste remineralised caries lesions [17]. The study also reported a great reduction in lesion depth, and formation of a new enamel layer was noticed using nanohydroxyapatite toothpaste. It reported that the combined effects of a nanohydroxyapatite and fluoride mouth rinse on an early caries lesion in human enamel improves remineralisation. They showed that the level of remineralisation was proportionate to the concentration of nanohydroxyapatite. In addition, yet another study reported that nanohydroxyapatite exhibits a synergistic role in remineralisation with a fluoride mouth rinse [18]. However, researchers should conduct further study to determine the optimum concentration of nanohydroxyapatite and sodium fluoride in mouth rinse for clinical applications.

A similar study reported that the microhardness decreased significantly after immersion in a demineralisation solution and increased following immersion in a nanohydroxyapatite and sodium fluoride mouth rinse [19]. Although this increase was not statistically significant, this study reported that nanohydroxyapatite and sodium fluoride mouth rinses enhance remineralisation and tooth microhardness. In a comparative study, researchers reported that nanohydroxyapatite gel can significantly remineralise enamel and cementum caries [20]. Another study demonstrated that nanohydroxyapatite can significantly increase microhardness of tooth enamel following exposure to soft drinks [21].

Furthermore, other studies concerned nanohydroxyapatite-incorporated dental materials. One such study found that nanohydroxyapatite-incorporated resin infiltrants can improve the prevention of recurrent demineralisation [22]. Moreover, another study showed that a fluorine-free toothpaste containing biomimetic nanohydroxyapatite has potential in preventing dental caries in the primary tooth [23]. This toothpaste prevents fluorosis and remineralizes and repairs enamel. The same study also reported that an acidic paste consisting of fluoride-hydroxyapatite was applied to repair small caries lesions. These researchers found that nanocrystalline growth can rapidly and seamlessly repair early caries with negligible wastage of enamel structure [29]. There was outstanding potential in using nanohydroxyapatite, but most of it was the in vitro stage. Researchers should conduct more advanced studies to validate the laboratory findings for the development of a novel nontoxic remedy that could be introduced in caries prevention.

Calcium phosphate nanoparticles can substitute for the lost hydroxyapatite by forming a new layer on carious teeth through the depositing of calcium and phosphate ions [107]. One study revealed that nanoamorphous calcium phosphate incorporated adhesive-enhanced dentin remineralisation [37]. These researchers reported that the incorporated adhesive can enhance acid neutralisation, thus enhancing calcium and phosphate content. In addition, it maintained a strong bond interface, inhibited secondary caries, and increased the longevity of the restoration. At the same time, some researchers reported on the amorphous calcium phosphate capability to have calcium and phosphate ions recharge and rerelease [38,108].

Simultaneously, the researchers described nanoamorphous calcium phosphate to provide long-term and sustained release of calcium and phosphate ions to create an anticaries environment. They also suggested using nanoamorphous calcium phosphate in conjunction with dental adhesives, composites, cement, and pit and fissure sealants to provide long-term anticaries properties. Researchers also studied amorphous calcium phosphate-containing orthodontic cement for effective caries inhibition and remineralisation to avoid white spot lesions in orthodontic treatments [36].

Nanoamorphous calcium phosphate with an antibacterial agent exhibited antibacterial and remineralising actions. The researchers reported nanoamorphous calcium phosphate-dimethylaminohexadecyl methacrylate composites as antibacterial and remineralising agents. Several studies have demonstrated that they inhibit lactic acid production, biofilm growth, and demineralisation. At the same time, they increase bond strength with dentin [42,43,44,46]. Moreover, another study reported that nanoamorphous calcium phosphate and quaternary ammonium methacrylate composites can inhibit oral microbes and their biofilm. They can also enhance the recovery of the dentin–pulp complex and dentin reformation [109].

In one study, 2-methacrylox-ylethyl dodecyl methyl ammonium bromide and nanoamorphous calcium phosphate exhibited antibacterial activities and remineralising properties without altering the bond strength of the composite resin [59]. Another study investigated the salivary statherin protein-inspired poly(amidoamine) dendrimer and adhesive containing nanoamorphous calcium phosphate in a cyclic artificial saliva/demineralising solution [49]. The nanocomposites exhibited significant remineralisation of artificial caries.

A nanoamorphous calcium phosphate and dimethylaminohexadecyl methacrylate nanocomposite-incorporated dental adhesive showed antibacterial and remineralisation capabilities. In addition, the nanocomposite inhibited biofilm without changing mechanical properties. Thus, it exhibited the ability to be incorporated into other existing dental materials [47]. Another study used nanoamorphous calcium phosphate and dimethylaminohexadecyl methacrylate nanocomposites with a resin-based crown cement. The nanocomposite showed the development of antibacterial activity against a saliva microcosm biofilm.

In addition, self-healing microcapsules (poly[urea-formaldehyde] shells containing triethylene glycol dimethacrylate, dimethylaminohexadecyl methacrylate, and nanoamorphous calcium phosphate) have been developed for preventing secondary caries. This agent has shown antibacterial and remineralising effects. The researchers demonstrated that this agent has excellent dentin bond strength, autonomous crack-healing and fracture toughness, and strong antibiofilm properties [60].

In a different study, a fluoride dentifrice containing nanocalcium phosphate remineralised early caries and prevented artificial incipient caries [50,51]. In yet another study, the researchers used nanocalcium carbonate for enamel remineralisation. The carbonate had the potential to remineralise incipient enamel caries using the unique anticaries properties of nanocalcium carbonate. In addition, these researchers reported that nanocalcium carbonate can be retained on oral surfaces and release calcium ions into oral fluids [48].

Two studies developed rechargeable agents that can provide long-term release of calcium and phosphate ions for remineralisation and reduction of caries [52,53]. In addition, there was no adverse effect on dentin bond strength. Nanoamorphous calcium phosphate-containing adhesives had high remineralising properties. The study demonstrated that nanoamorphous calcium phosphate adhesive released calcium and phosphate, neutralised acidic conditions, and reduced the production of lactic acid and biofilm. The developed adhesive also has potential for remineralisation [41]. Another study used nanosilver, quaternary ammonium dimethacrylate, and nanoamorphous calcium phosphate with adhesive. They suggested that this novel approach of combining antimicrobial and remineralising agents with adhesive could be used for caries prevention [56].

One study demonstrated that calcium phosphate nanoparticle-filled dental cement showed good bond strength to enamel, calcium, and phosphate ion recharge/rerelease. The cement can also inhibit biofilm to reduce caries [54]. Other researchers developed calcium fluoride nanoparticles incorporated in a nanocomposite. They reported that nanocomposites have high fluoride release, strong mechanical properties, durability, high strength, and high load-bearing capacities. Calcium fluoride nanocomposites could be a promising stress-bearing and caries-inhibiting restorative material [31].

One study showed that calcium fluoride nanoparticles reduced biofilm formation and exopolysaccharide production. The same study also reported that calcium fluoride nanoparticles substantially inhibit cariogenic biofilm and could be used as a topical anticaries agent [32]. Another study suggested that calcium fluoride nanoparticles enhanced remineralisation by increasing labile fluoride concentration in the oral fluid [30]. Still another study incorporated calcium fluoride nanoparticles with dimethylaminohexadecyl methacrylate to increase the release of fluoride and calcium ions to promote remineralisation [33,34].

Besides the antibacterial effect, the current research focuses on synthesising biomimetic dental enamel using calcium phosphate nanoparticles as a widely accepted research theme. One study investigated the biomimetic remineralisation potential of a calcium phosphate polymer-induced liquid precursor at demineralised artificial caries and dentin caries lesions. They have shown biomimetic remineralisation with better bonding of interfacial of the biomimetic remineralised artificial caries dentin lesion [57].

Another study showed that phosphorylated chitosan–amorphous calcium phosphate exhibited biomineralisation to form a dental hard tissue-like structure that resembles enamel structures. In addition, the remineralisation of enamel by using phosphorylated chitosan–amorphous calcium phosphate was higher than that of fluoride [58]. Researchers should conduct advanced studies to translate these potentials in in vivo and clinical settings.

### 4.3. Bioactive Glass-Based Nanoparticles

Bioactive glasses are surface-reactive bioceramic materials widely used in biomedical applications [110]. They usually dissolve in body fluids and form apatite crystals that resemble bone and tooth tissue, and exhibit chemical bonds to the bone and tooth surface [111,112]. In one study, nanobioactive glass containing resin composites showed a uniform apatite layer formation on the tooth surface with no negative effects on their underlying properties [113].

At the same time, a nanobioactive glass-containing composite increased the microhardness of demineralised dentin [114]. Nanobioactive glass can induce hydroxyapatite formation and osteoinductive ability. In oral conditions, nanobioactive glass forms hydroxyapatite on the dentin surface, seals the orifices of the dentinal tubules, and reduces dentin permeability and sensitivity [115,116,117]. Demineralised dentin treated with nanobioactive glass was enriched with minerals and ions. The nanobioactive glass also increased the microhardness of the carious lesion surface [118,119].

Although remineralisation is well-reported, researchers have not well explored its mechanism. There are no clear data on whether it is intrafibrillar or extrafibrillar mineralisation. In one study, arginine–glycine–aspartate–serine was conjugated with nanobioactive glass. The conjugate showed a crystal lattice formation in the demineralised dentin matrix. The crystal lattice has the highest dentin cohesive strength and intrafibrillar mineralisation. Thus, researchers have suggested using this composite in dentin erosion, hypersensitivity, a bonding interface, and regenerative dentistry [62].

Simultaneously, a study demonstrated that nanobioactive glass can increase the Vickers hardness number and reduce caries depth on the surface of the lesion [63]. Interestingly, researchers have found that dental materials with bioactive glass can release ions to inhibit dental caries. The study showed nanobioactive glass contents in sealant-enhanced inhibition of demineralisation of the enamel surface in a cariogenic environment. The researchers stated that despite some marginal leakage, these novel sealants were effective in inhibiting secondary caries at the margins [64].

On the other hand, researchers have also reported that nanobioactive glass apatite can induce cell proliferation and differentiation of dental stem cells into a mineralising lineage. One study investigated the effects of nanobioactive glass on the odontogenic differentiation and mineralisation of human dental pulp cells. The nanobioactive glass can exhibit enhanced alkaline phosphatase activity, collagen type I, dentin sialophosphoprotein, dentin matrix protein 1 production, and mineralised nodule formation [66].

Similarly, nanobioactive glass can also induce odontogenic differentiation of rat dental pulp stem cells, which might be used as a potential dentin regenerative additive for existing or new dental material for enhancing odontoblast differentiation [67]. An investigation explored whether nanobioactive glass-incorporated endodontic sealer promoted cementoblast differentiation of human periodontal ligament stem cells without any growth factors. It also enhanced gene expression for the production of mineralised tissues [68].

Studies have also reported the antimicrobial and remineralisation properties of nanobioactive glass [69,73]. One study reported that nanobioactive glass has a stronger antibacterial and antibiofilms effect than that of triclosan or sodium fluoride alone. What is more, nanobioactive glass combined with either triclosan or sodium fluoride may exert an addictive antibacterial effect and enhanced biofilm inhibition effect [82].

Combining apatite-forming capability and antimicrobial activity, nanobioactive glass was incorporated into several commercial dental products, especially in toothpaste [120,121]. Studies showed that when nanobioactive glass-incorporated toothpaste is introduced into the oral environment, the toothpaste can release sodium, calcium, and phosphate ions. These ions react with oral fluid and form crystalline hydroxyapatite that structurally and chemically resembles tooth minerals. The nanobioactive glass can increase remineralisation and seal dentinal tubules. It can also provide continuous occlusion and inhibit tooth sensitivity. Thus, it can potentially be used for remineralisation and caries prevention [70,71,115].

Nanobioactive glass is capable of depositing layers of hydroxyl carbonate apatite in body fluids. In one study, researchers treated caries lesions in human dental enamel with a nanobioactive glass paste and phosphoric acid. This paste formed a crystalline layer that was later converted to hydroxyapatite crystals in artificial saliva. The researchers suggested restoring incipient enamel erosive lesions with an abrasion-durable layer of hydroxyapatite crystals [65]. At the same time, some researchers reported that fluoride- and phosphate-incorporated nanobioactive glass can form fluorapatite, which is more active than hydroxyapatite in resisting an acidic environment. In addition, they can provide dentin sealing and control the release of calcium, phosphate, and fluoride ions for a longer period after tooth brushing [74,75].

Chitosan–nanobioactive glass is effective in promoting subsurface mineral deposition without the salivary pellicle. It exhibits greater mineral deposition and enhanced subsurface microhardness. Therefore, the researchers reported that it is promising for remineralising enamel caries as well as desensitising exposed porous vital dental tissues. Thus, the researchers suggested using it as an alternative clinical strategy in caries prevention [81]. A study reported that nanobioactive glass containing resin bonding can reduce microleakages of the resin–dentin interface by depositing minerals that facilitate remineralisation [72].

On the other hand, it has been shown that silver-doped bioactive glass/mesoporous silica nanoparticles can effectively seal the orifices of the dentinal tubules in acidic conditions and form a membrane-like layer. Moreover, it did not decrease bond strength in the self-etch adhesive system and had low or negligible cytotoxicity and antibacterial effects [83]. Some researchers incorporated nanobioactive glass into Biodentine^TM^ to form nanobioactive glass–biodentine composites. The product can accelerate apatite formation, seal the orifices of the dentinal tubules, and enhance the formation of a mineral-rich interfacial layer on the dentine surface [76]. Demineralised enamel and dentin surfaces treated with nanobioactive glass and amorphous calcium phosphate–casein phosphopeptide showed a highly significant increase in microhardness. They effectively remineralised the early caries of enamel. However, nanobioactive glass showed better results initially, but eventually both had a similar remineralising potential [78,79].

In another study, researchers found that nanobioactive glass powder and nanobioactive glass containing polyacrylic acid could enhance the remineralisation of enamel of white spot lesions. The material also exhibited significantly higher surface and cross-section Knoop microhardness. Although there was a significant mineral deposition, lesion depth was not significantly reduced [80]. Another study evaluated nanobioactive glass ceramic for erosion and caries control. The ceramic can exhibit a higher potential in reducing surface loss and initiation and the progression of erosion and enamel caries [77]. Nanobioactive glass ceramics are opening a noninvasive treatment strategy for caries prevention. More advanced research will find the optimal application of nanobioactive glass in caries management.

### 4.4. Silica-Based Nanoparticles

Silica is an inorganic ceramic material composed of silicon dioxide [122]. Silica in a colloidal solution occurs as an insoluble dispersion of amorphous fine silica particles [123]. Silica is one of the attractive minerals for collagen infiltration. It is assumed that it can penetrate the demineralised collagen matrix without precipitating on the surface [116,124]. Calcium-doped mesoporous silica nanoparticles as inorganic fillers improve the mechanical properties of the resin composites. Some researchers have suggested using these nanoparticles as a carrier for ciprofloxacin hydrochloride loading to add antibacterial properties to facilitate secondary caries prevention [84].

In one study, calcium mesoporous silica nanoparticles were shown to reduce roughness and to be effective in minimising tooth surface loss compared to that of casein phosphopeptide–amorphous calcium phosphate, titanium fluoride, and sodium fluoride. Therefore, the researchers suggested that these nanoparticles are promising in reducing dental erosion [35]. In other studies, researchers have studied nanohydroxyapatite and silica nanoparticles on erosive enamel and dentin lesions. They reported that the mineral deposition in enamel was not statistically different. However, in dentin, nanohydroxyapatite infiltrated significantly more minerals than did the nanosilica infiltrant [85].

Some researchers have investigated a versatile dentin surface biobarrier comprising a mesoporous silica-based epigallocatechin-3-gallate/nanohydroxyapatite delivery system. This system can protect orifices of the dentinal tubules against acid and abrasion, reduce dentin permeability, and inhibit the *S. mutans* biofilm formation to protect the exposed dentin [92]. In another study, demineralised dentin infiltrated with silica nanoparticles exhibited enhanced heterogeneous mineralisation of the dentin collagen matrix in an artificial saliva solution [86]. Another study showed that bioactive tricalcium silicate was capable of repairing the acid-etched enamel. Thus, the researchers suggested it as a potential in protecting demineralised teeth [88]. Similarly, other researchers also reported that tricalcium silicate paste may have the potential for remineralising subsurface enamel lesions [89].

Some researchers have reported that orthodontic adhesives containing calcium silicate are effective for acid neutralisation, apatite formation, and enamel remineralisation [90]. One study showed that collagen infiltrated with hydroxyapatite and nanosilica can be used as a scaffold for remineralising dentin [87]. Another study used rice husk nanosilica and demonstrated that they exhibit dentin hydroxyapatite formation. In addition, they exhibited antimicrobial effects [91]. On the other hand, another study reported that mesoporous silica biomaterials had the potential to be a catalyst and carrier in the repair and/or regeneration of dental hard tissue [94].

To incorporate antimicrobial activity into glass ionomer cement without altering its mechanical properties, some researchers have added mesoporous silica nanoparticles-encapsulated chlorhexidine to glass ionomer cement. These researchers found that the nanoparticles may obtain antibiofilm ability with no adverse effects on mechanical properties. Thus, mesoporous silica nanomaterials can be suggested as a new strategy for preventing secondary caries [93]. Researchers should conduct further studies to choose the better application of silica nanoparticles in caries prevention as well as in clinical dentistry.

## 5. Conclusions

In conclusion, ceramic minerals have potential in the prevention of dental caries. They facilitate the remineralisation process and maintain the equilibrium of pH levels to maintain tooth integrity. Antimicrobial nanoparticles-conjugated ceramic minerals provide dual action and prevent caries formation and progression.

## Figures and Tables

**Figure 1 nanomaterials-12-04416-f001:**
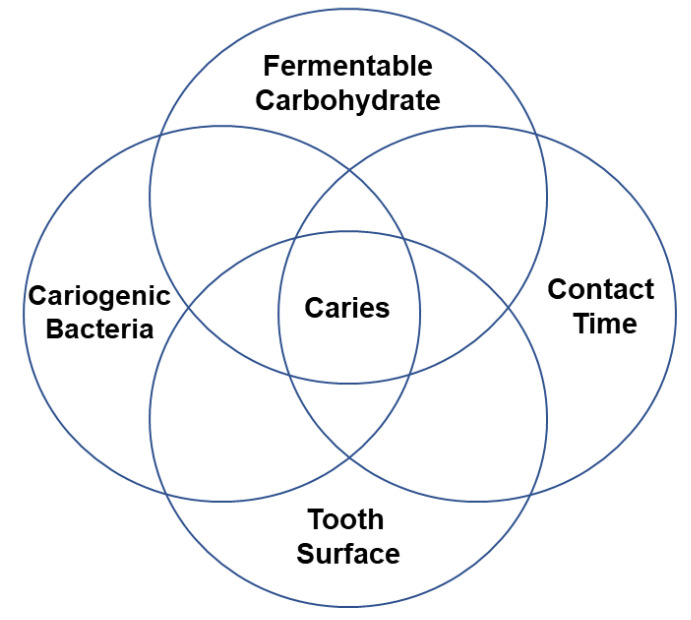
Schematic illustration of the initiation of caries formation [8].

**Figure 2 nanomaterials-12-04416-f002:**
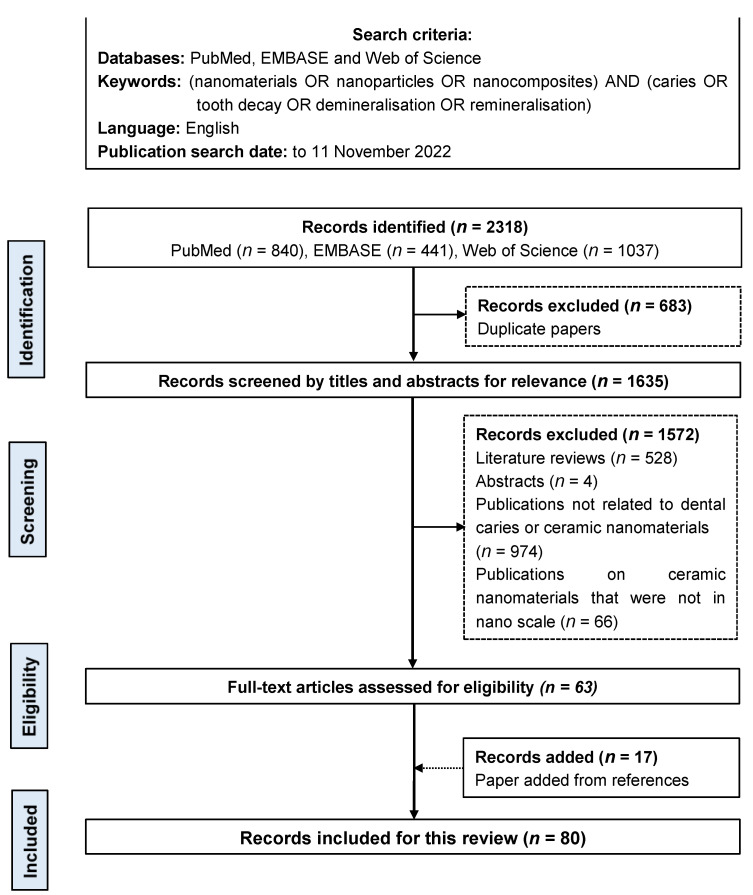
Flow chart of the literature search.

**Figure 3 nanomaterials-12-04416-f003:**
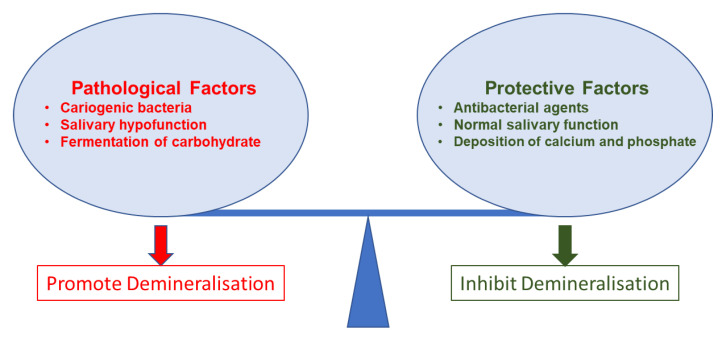
Pathological and protective factors affecting demineralisation of tooth tissue.

**Figure 4 nanomaterials-12-04416-f004:**
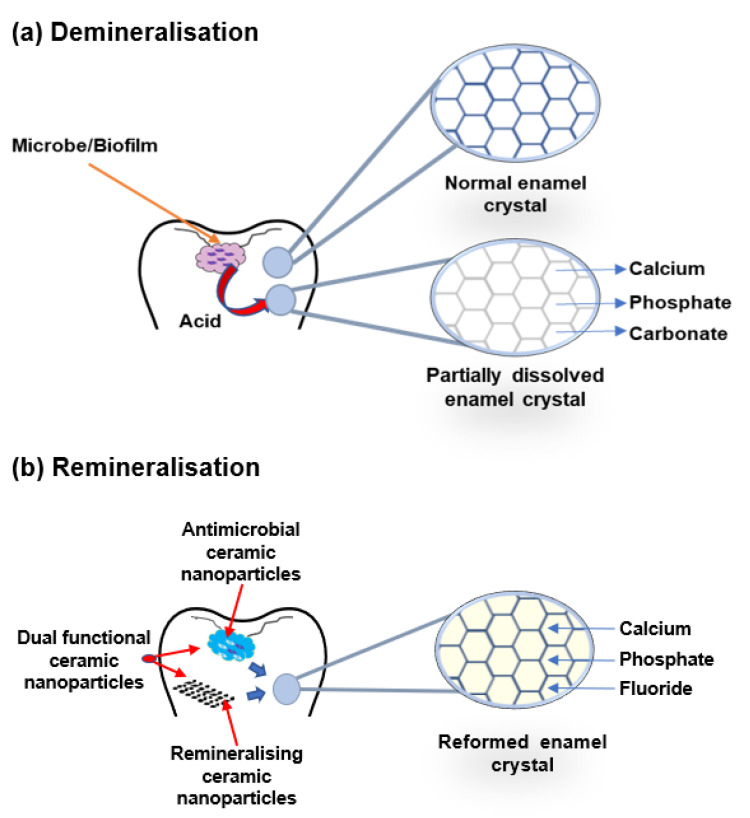
Schematic illustration of enamel demineralisation and remineralisation. (**a**) Demineralisation—acid from biofilm dissolves the enamel crystal and leaks out minerals such as calcium, phosphate, and carbonate. (**b**) Remineralisation—antimicrobial or remineralising or dual functional ceramic nanoparticles inhibit cariogenic biofilm and restore the lost minerals by the accumulation of calcium, phosphate, and fluoride in the partially dissolved enamel crystal.

**Table 1 nanomaterials-12-04416-t001:** Types of ceramic materials with anticaries properties.

Types of Ceramic Materials	Anticaries Properties [Reference(s)]
** *1. Calcium-Based Nanoparticles* **	
Nanohydroxyapatite	[11,12,13,14,15,16,17,18,19,20,21,22,23,24,25,26,27,28]
Nanohydroxyapatite (synthetic enamel)	[29]
Calcium fluoride nanoparticles	[30,31,32]
Calcium fluoride nanoparticles and dimethylaminohexadecyl methacrylate	[33,34]
Casein phosphopeptide-amorphous calcium phosphate	[35]
Nanoamorphous calcium phosphate	[36,37,38,39,40,41]
Dimethylaminohexadecyl methacrylate and nanoamorphous calcium phosphate	[42,43,44,45,46,47]
Calcium carbonate	[48]
Statherin protein-inspired poly(amidoamine) dendrimer and nanoamorphous calcium phosphate	[49]
Nanocalcium phosphate	[36,42,43,50,51,52,53,54,55,56]
Calcium phosphate polymer-induced liquid precursor	[57]
Phosphorylated chitosan-amorphous calcium phosphate	[58]
2-methacrylox-ylethyl dodecyl methyl ammonium bromide and nanoamorphous calcium phosphate	[59]
Triethylene glycol dimethacrylate, dimethylaminohexadecyl methacrylate, and nanoamorphous calcium phosphate	[60,61]
** *2. Bioactive Glass-Based Nanoparticles* **	
Nanobioactive glass and arginine-glycine-aspartate-serine	[62]
Nanobioactive glass	[63,64,65,66,67,68,69,70,71,72,73]
Nanobioactive glass and fluoride	[74,75]
Nanobioactive glass and biosilicate	[76,77]
Nanobioactive glass and amorphous calcium phosphate-casein phosphopeptide	[78,79]
Nanobioactive glass and polyacrylic acid	[80]
Nanobioactive glass and chitosan	[81]
Nanobioactive glass, sodium fluoride, and triclosan	[82]
Nanobioactive glass, silver, and silica	[83]
** *3. Silica-Based Nanoparticles* **	
Mesoporous silica nanoparticles and calcium	[35,84]
Silica nanoparticles and nanohydroxyapatite	[85,86,87]
Tricalcium silicate	[88,89,90]
Nano-silica	[91]
Mesoporous silica-based epigallocatechin-3-gallate and nanohydroxyapatite	[92]
Mesoporous silica nanoparticles-encapsulated chlorhexidine	[93,94]

## Data Availability

Not applicable.

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
