# Peer review of "Ceramic Nanomaterials in Caries Prevention: A Narrative Review"

_nanomaterials, 2022, doi:10.3390/nano12244416_

Round 1
Reviewer 1 Report
The paper wishes to offer a good insight on the state of the art in the field of ceramic nanomaterials for dental repair. However, there is a need of careful rewriting of the paper, since some information originally presented in the quoted papers appears a bit distorted in this review. The authors should carefully re-read those papers, have a clear understanding of the methods, results and discussion displayed and then re-systematize their exposure. No doubt, the intention is good, but, however, for the moment the outcome is not fully satisfactory.
The paper is in need of a deep revision, I just provide a few examples:
In thable 1, Antibacterial action – refs. 25, 26, there is no such mantion in the cited papers to this activity.
The authors mention the use of adhesive(s) in dental repair compositions, such as in Lines 267-70: The mention af adhesive (which may represent another entity) is present, as in other several instances, but no mention of the nature of such adhesive is made. There should be a presentation of the adhesive(s) used. These could have properties that may somewhat interfere with the reconstruction.
At L 293 the authors claim “Another study showed that phosphorylated chitosan-amorphous calcium phosphate exhibited biomineralisation to form a dental hard tissue-like structure that resembles enamel structures. In addition, the remineralisation of enamel was higher than that of fluoride”
It is an unclear phrase… how the remineralization of enamel was harder than that of fluoride? They should be mor clear in the exprimation.
L314 In one study, nano-bioactive glass-induced arginine-glycine-aspartate-serine showed a crystal lattice formation in the demineralised dentin matrix; I cannot understand how the bioactive glass was induced by (or induced) arg-gli-asp-ser? The authors should rephrase to make this understandable. According to the authors of the cited article, the Arg-gli-asp-ser peptide has the ability to associate to dentin, and thus promote the binding of nanobioactive glass…
The cited paper is saying that this peptide has the ability to promote the biding of biaoactve lass to dentin, which is a bit different from the authors interpretation.
Ince there is not only a language error, but, in my opinion, an incomplete or a misunderstanding of the cited papers, I suggest a revision of the paper.
Author Response
General Comments: The paper wishes to offer a good insight on the state of the art in the field of ceramic nanomaterials for dental repair. However, there is a need of careful rewriting of the paper, since some information originally presented in the quoted papers appears a bit distorted in this review. The authors should carefully re-read those papers, have a clear understanding of the methods, results and discussion displayed and then re-systematize their exposure. No doubt, the intention is good, but, however, for the moment the outcome is not fully satisfactory.
The paper is in need of a deep revision, I just provide a few examples:
In thable 1, Antibacterial action – refs. 25, 26, there is no such mantion in the cited papers to this activity.
Response: Thank you very much for your time and effort in reviewing our review. We re-read our quoted papers and checked our citations. We have revised Table 1. accordingly to avoid unwanted confusion.
Comments 1: The authors mention the use of adhesive(s) in dental repair compositions, such as in Lines 267-70: The mention af adhesive (which may represent another entity) is present, as in other several instances, but no mention of the nature of such adhesive is made. There should be a presentation of the adhesive(s) used. These could have properties that may somewhat interfere with the reconstruction.
Response: Thank you for your comment. We have revised the manuscript and highlighted them in yellow (lines 291-292).
Comments 2: At L 293 the authors claim “Another study showed that phosphorylated chitosan-amorphous calcium phosphate exhibited biomineralisation to form a dental hard tissue-like structure that resembles enamel structures. In addition, the remineralisation of enamel was higher than that of fluoride”
It is an unclear phrase… how the remineralization of enamel was harder than that of fluoride? They should be mor clear in the exprimation.
Response: We have revised the manuscript to make it clear. The changes are highlighted in yellow in the main manuscripts (lines 316-317).
Comments 3: L314 In one study, nano-bioactive glass-induced arginine-glycine-aspartate-serine showed a crystal lattice formation in the demineralised dentin matrix; I cannot understand how the bioactive glass was induced by (or induced) arg-gli-asp-ser? The authors should rephrase to make this understandable. According to the authors of the cited article, the Arg-gli-asp-ser peptide has the ability to associate to dentin, and thus promote the binding of nanobioactive glass…
The cited paper is saying that this peptide has the ability to promote the biding of biaoactve lass to dentin, which is a bit different from the authors interpretation.
Response: We have revised the manuscript to make it clear. The changes are highlighted in yellow (lines 338-339).
Comments 4: Ince there is not only a language error, but, in my opinion, an incomplete or a misunderstanding of the cited papers, I suggest a revision of the paper.
Response: We have revised our manuscript accordingly to make it clear, consistent, and understandable.

Reviewer 2 Report
Dear Authors,
I have read the manuscript with interest and some questions raised. Enlisted please find my comments.
Overall. General English grammar revision (Minor spelling errors).
Key words. “dentistry” and “prevention” could be added in my opinion.
Introduction. Authors stated “However, caries is a preventable disease, and early caries can be remineralised under a favourable environment.”. Please add a reference for this statement.
Figure 1. Please add a reference for the mentioned Schematic illustration of the initiation of caries formation.
Introduction. Authors stated “Inhibiting cariogenic bacteria and biofilm or enhancing remineralisation, or applying dual action can be a scientific approach for preventing the initiation of primary caries”. After this sentence an overview of remineralizing materials should be given. It could be stated that “fluoride has been the first attempt in dental practice used for preventive purposes (Historical and bibliometric notes on the use of fluoride in caries prevention. Zampetti P, Scribante A. Eur J Paediatr Dent. 2020 Jun;21(2):148-152.). Subsequently, casein phosphopeptide-amorphous calcium phosphate (Quantitative evaluation of remineralizing potential of three agents on artificially demineralized human enamel using scanning electron microscopy imaging and energy-dispersive analytical X-ray element analysis: An in vitro study. Khanduri N, Kurup D, Mitra M. Dent Res J (Isfahan). 2020 Sep 7;17(5):366-372.) hs been recently introduced and showed promising results”. These concerns should be added before the sentence “Nowadays Researchers are using nanotechnology to develop multifunctional nanomaterials for preventing caries”.
Introduction. Authors started “Ceramic nanomaterials are nano-scale materials that are inorganic metalloid solids made up of oxides, carbides, carbonates, and phosphates. These ceramic nanomaterials are synthesized by heating at high temperatures followed by rapid cooling. Ceramic nanoparticles are more stable and cheaper in production than metallic nanoparticles”. Please add a reference for this statement.
Literature survey. Please clearly state the mesh terms used.
Methods. Please add a table showing the risk of bias with a quality assessment of the main studies taken into account to perform revision. Please use the three codified colors to identify low (green), moderate (yellow) or high (red) risk of bias. Please add details about how this information is to be used in any data synthesis.
Results. Give numbers of studies screened, assessed for eligibility, and included in the review, with reasons for exclusions at each stage, ideally with a flow diagram.
Discussion. Discuss limitations at study and outcome level (e.g., risk of bias), and at review-level (e.g., incomplete retrieval of identified research, reporting bias).
Discussion. Provide a general interpretation of the results in the context of other evidence, and implications for future research.
Author Response
General comments: Dear Authors,
I have read the manuscript with interest and some questions raised. Enlisted please find my comments.
Overall. General English grammar revision (Minor spelling errors).
Our response: Thank you so much for your appreciation.
Comments 1: Key words. “dentistry” and “prevention” could be added in my opinion.
Response: We have added those keywords in the main manuscript and highlighted them in yellow (lines 32-33).
Comments 2: Introduction. Authors stated “However, caries is a preventable disease, and early caries can be remineralised under a favourable environment.”. Please add a reference for this statement.
Response: We have cited a reference (ref 2) in the main manuscript and highlighted it in yellow (line 40).
Comments 3: Figure 1. Please add a reference for the mentioned Schematic illustration of the initiation of caries formation.
Response: We have cited a reference (ref 8) in the main manuscript and highlighted it in yellow (line 57).
Comments 4: Introduction. Authors stated “Inhibiting cariogenic bacteria and biofilm or enhancing remineralisation, or applying dual action can be a scientific approach for preventing the initiation of primary caries”. After this sentence an overview of remineralizing materials should be given. It could be stated that “fluoride has been the first attempt in dental practice used for preventive purposes (Historical and bibliometric notes on the use of fluoride in caries prevention. Zampetti P, Scribante A. Eur J Paediatr Dent. 2020 Jun;21(2):148-152.). Subsequently, casein phosphopeptide-amorphous calcium phosphate (Quantitative evaluation of remineralizing potential of three agents on artificially demineralized human enamel using scanning electron microscopy imaging and energy-dispersive analytical X-ray element analysis: An in vitro study. Khanduri N, Kurup D, Mitra M. Dent Res J (Isfahan). 2020 Sep 7;17(5):366-372.) hs been recently introduced and showed promising results”. These concerns should be added before the sentence “Nowadays Researchers are using nanotechnology to develop multifunctional nanomaterials for preventing caries”.
Response: We have added the recommended lines and cited 2 references (ref 3 and 4) in the main manuscript and highlighted them in yellow (lines 42-45).
Comments 5: Introduction. Authors started “Ceramic nanomaterials are nano-scale materials that are inorganic metalloid solids made up of oxides, carbides, carbonates, and phosphates. These ceramic nanomaterials are synthesized by heating at high temperatures followed by rapid cooling. Ceramic nanoparticles are more stable and cheaper in production than metallic nanoparticles”. Please add a reference for this statement.
Response: We have cited a reference (ref 9) in the main manuscript and highlighted it in yellow (line 60).
Comments 6: Literature survey. Please clearly state the mesh terms used.
Methods. Please add a table showing the risk of bias with a quality assessment of the main studies taken into account to perform revision. Please use the three codified colors to identify low (green), moderate (yellow) or high (red) risk of bias. Please add details about how this information is to be used in any data synthesis.
Results. Give numbers of studies screened, assessed for eligibility, and included in the review, with reasons for exclusions at each stage, ideally with a flow diagram.
Discussion. Discuss limitations at study and outcome level (e.g., risk of bias), and at review-level (e.g., incomplete retrieval of identified research, reporting bias).
Discussion. Provide a general interpretation of the results in the context of other evidence, and implications for future research.
Response: We understood the reviewer’s concern. We did a systematic search. We have added the flowchart of search results in the main manuscripts and highlighted them in yellow as suggested. To avoid confusion we have rewritten the “Literature survey” section and revised the structure of the review in the main manuscript and highlighted them in yellow (lines 72-97).
We do not want to judge the quality of peer reviewed articles. Because this study is a narrative review, we did not intend to a systematic review to assess the risk of bias and quality assessment of the literatures.

Round 2
Reviewer 1 Report
Your so called revised version still contains invented text, never present in the quoted papers, to support your affirmations. Putting words in a paper that were not present is a major ethical misconduct. Therefore I suggest rejection of your paper.
Reviewer 2 Report
All comments have been assessed